# Learning Robust Dynamics through Variational Sparse Gating

**Arnav Kumar Jain**[1,2,*], **Shivakanth Sujit**[2,3], **Shruti Joshi**[2,3], **Vincent Michalski**[1,2]
**Danijar Hafner**[4,5], **Samira Ebrahimi-Kahou**[2,3,6]

## Abstract

Learning world models from their sensory inputs enables agents to plan for actions by imagining their future outcomes. World models have previously been shown to improve sample-efficiency in simulated environments with few objects, but have not yet been applied successfully to environments with many objects. In environments with many objects, often only a small number of them are moving or interacting at the same time. In this paper, we investigate integrating this inductive bias of sparse interactions into the latent dynamics of world models trained from pixels. First, we introduce Variational Sparse Gating (VSG), a latent dynamics model that updates its feature dimensions sparsely through stochastic binary gates. Moreover, we propose a simplified architecture Simple Variational Sparse Gating (SVSG) that removes the deterministic pathway of previous models, resulting in a fully stochastic transition function that leverages the VSG mechanism. We evaluate the two model architectures in the BringBackShapes (BBS) environment that features a large number of moving objects and partial observability, demonstrating clear improvements over prior models.

## 1 Introduction

Latent dynamics models are models that generate agent's future states in the compact latent space without feeding the high-dimensional observations back to the model. They have shown promising results on various tasks like video prediction (Karl et al., 2016; Kalman, 1960; Krishnan et al., 2015), model-based Reinforcement Learning (RL) (Hafner et al., 2020; 2021; 2019; Ha and Schmidhuber, 2018), and robotics (Watter et al., 2015). Generating sequences in the compact latent space reduces the accumulating errors leading to more accurate long-term predictions. Additionally, having lower dimensionality leads to a lower memory footprint. Solving tasks in model-based RL involves learning a world model (Ha and Schmidhuber, 2018) that can predict outcomes of actions, followed by using them to derive behaviors (Sutton, 1991). Motivated by these benefits, the recently proposed DreamerV1 (Hafner et al., 2020) and DreamerV2 (Hafner et al., 2021) agents achieved state-of-the-art results on a wide range of visual control tasks.

Many complex tasks require reliable long-term prediction of dynamics. This is true especially in partially observable environments where only a subspace is visible to the agent, and it is usually required to accurately retain information over multiple time steps to solve the task. The Dreamer agents (Hafner et al., 2020; 2021) employ an Recurrent State-Space Model (RSSM) (Hafner et al., 2019) comprising of a Recurrent Neural Network (RNN). Training RNNs for long sequences is challenging as they suffer from optimization problems like vanishing gradients (Hochreiter, 1991; Bengio et al., 1994). Different ways of applying sparse updates in RNNs have been investigated (Campos et al., 2017; Neil et al., 2016; Goyal et al., 2019), enabling a subset of state dimensions to be constant

36th Conference on Neural Information Processing Systems (NeurIPS 2022).

[1]Université de Montréal, [2]Mila- Quebec AI Institute, [3]École de technologie supérieure, [4]University of Toronto, [5]Google Brain, [6]CIFAR. [*]Correspondence to Arnav Kumar Jain <arnav-kumar.jain@mila.quebec>.

during the update. A sparse update prior can also be motivated by the fact that in the real world, many factors of variation are constant over extended periods of time. For instance, several objects in a physical simulation may be stationary until some force acts upon them. Additionally, this is useful in the partially observable setting where the agent observes a constrained viewpoint and has to keep track of objects that are not visible for many time steps. In this work, we introduce Variational Sparse Gating (VSG), a stochastic gating mechanism that sparsely updates the latent states at each step.

Recurrent State-Space Model (RSSM) (Hafner et al., 2019) was introduced in PLaNet where the model state was composed of two paths, an image representation path and a recurrent path. DreamerV1 (Hafner et al., 2020) and DreamerV2 (Hafner et al., 2021) utilized them to achieve state-of-the-art results in continuous and discrete control tasks (Hafner et al., 2019). While the image representation path which is stochastic accounts for multiple possible future states, the recurrent path is deterministic to retain information over multiple time steps to facilitate gradient-based optimization. (Hafner et al., 2019) showed that both components were important for solving tasks, where the stochastic part was more important to account for partial observability of the initial states. By leveraging the proposed gating mechanism (Variational Sparse Gating (VSG)), we demonstrate that a purely stochastic model with a single component can achieve competitive results, and call it Simple Variational Sparse Gating (SVSG). To the best of our knowledge, this is the first work that shows that purely stochastic models achieve competitive performance on continuous control tasks when compared to leading agents.

Existing benchmarks (Bellemare et al., 2013; Chevalier-Boisvert et al., 2018; Tassa et al., 2018) for RL do not test the capability of agents in both partial observability and stochasticity. The Atari (Bellemare et al., 2013) benchmark comprises of 55 games but most of the games are deterministic and a lot of compute is required to train on them. Some tasks in the Atari and Minigrid benchmarks are partially-observable but either lack stochasticity or are hard exploration tasks. Also, these benchmarks do not allow for controlling the factors of variation. We developed a new partially-observable and stochastic environment, called BringBackShapes (BBS), where the task is to push objects to a predefined goal area. Solving tasks in BBS require agents to remember states of previously observed objects and avoid noisy distractor objects. Furthermore, VSG and SVSG outperformed leading model-based and model-free baselines. We also present studies with varying partial-observability and stochasticity to demonstrate that the proposed agents have better memory for tracking observed objects and are more robust to increasing levels of noise. Lastly, the proposed methods were also evaluated on existing benchmarks - DeepMind Control (DMC) (Tassa et al., 2018), DMC with Natural Background (Zhang et al., 2021; Nguyen et al., 2021b), and Atari (Bellemare et al., 2013). On the existing benchmarks, the proposed method performed better on tasks with changing viewpoints and sparse rewards.

Our key contributions are summarized as follows:

- **Variational Sparse Gating**: We introduce Variational Sparse Gating (VSG), where the recurrent states are sparsely updated through a stochastic gating mechanism. A comprehensive empirical evaluation shows that VSG outperforms baselines on tasks requiring long-term memory.
- **Simple Variational Sparse Gating**: We also propose Simple Variational Sparse Gating (SVSG) which has a purely stochastic state, and achieves competitive results on continuous control tasks when compared with agents that also use a deterministic component.
- **BringBackShapes**: We developed the BringBackShapes (BBS) environment to evaluate agents on partially-observable and stochastic settings where these variations can be controlled. Our experiments show that the proposed agents are more robust to such variations.

## 2  Variational Sparse Gating

**Reinforcement Learning**: The visual control task can be formulated as a Partially Observable Markov Decision Process (POMDP) with discrete time steps $t \in [1; T]$. The agent selects action $a_t \sim p(a_t | o_{\leq t}, a_{<t})$ to interact with the environment and receives the next observation and scalar reward $o_t, r_t \sim p(o_t, r_t | o_{<t}, r_{<t})$, respectively, at each time step. The goal is to learn a policy that maximizes the expected discounted sum of rewards $\mathbb{E}_p(\sum_{t=1}^{T} \gamma^t r_t)$, where $\gamma$ is the discount factor.

**Agent**: Agent is composed of a world model and a policy (Fig. 1). World models (Sec. 2.1) encode a sequence of observations and actions into latent representations. The agents behavior (Appendix B) is derived to maximize expected returns on the trajectories generated from the learned world model. While training, the world model is learned with collected experience, the policy is improved on

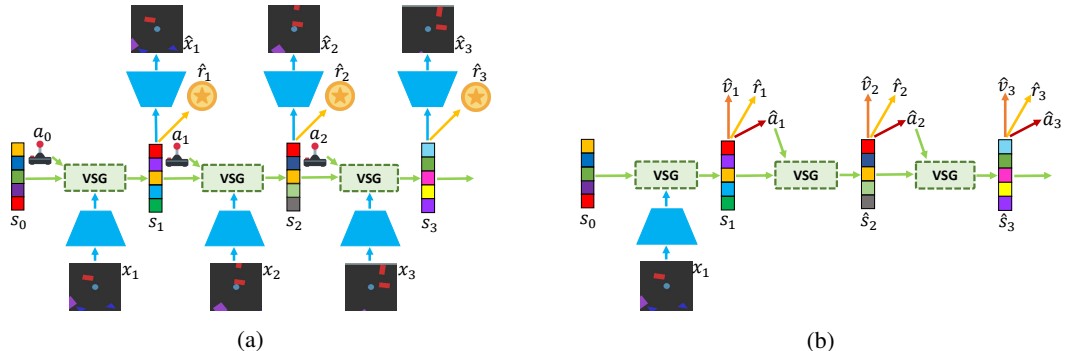

(a)                                  (b)

Figure 1: (a) World Model: The VSG block takes the previous model state $s_{t-1}$ and action $a_{t-1}$, and outputs the updated model state at next step $s_t$, which is further used to reconstruct image $\hat{x}_t$ and reward $\hat{r}_t$. (b) Policy: Comprises of an actor to select optimal action $\hat{a}_t$ and critic to predict value $\hat{v}_t$ beyond the planning horizon. The world model is unrolled using the prior model state $\hat{s}_t$ which does not contain information about image $x_t$.

trajectories unrolled using the world model and new episodes are collected by deploying the policy in the environment. An initial set of episodes are collected using a random policy. As training progresses, new episodes are collected using the latest policy to further improve the world model.

## 2.1 World Model

World Models (Ha and Schmidhuber, 2018) learn to mimic the environment using the collected experience and facilitate deriving behaviours in the abstract latent space. Given an abstract state of the world and an action, the model applies the learned transition dynamics to predict the resulting next state and reward. RSSM (Hafner et al., 2019) was introduced in PlaNet, where the model state was composed of two paths. The recurrent path consists of an RNN (See Figure 2 [a]), and is motivated with reliable long-term information preservation, while the image representation path samples from a learned distribution to account for multiple possible futures (Babaeizadeh et al., 2017). In this work, we introduce Variational Sparse Gating (VSG), where the recurrent path selectively updates a subset of the latent states at each step using a stochastic gating network. Sparse updates enable the agent to have long-term memory and learn robust representations to solve complex tasks.

**Model Components**: The world model comprises of an image encoder, a VSG model, and predictors for image, discount and reward. The image encoder generates representations $o_t$ for the observation $x_t$ using Convolutional Neural Networks (CNNs). The VSG model comprises of a recurrent model equipped with the stochastic gating mechanism to get the recurrent state $h_t$, and is used to compute two stochastic image representation states. The posterior representation state $z_t$ is obtained using the representation model and contains information about the current observation $x_t$. The prior state $\hat{z}_t$ is obtained from the transition predictor without observing the current observation $x_t$. This is useful while planning as sequences are generated in compact latent state, and the output from the transition predictor is utilized. This also results in a lower memory footprint and enables predictions of thousands of trajectories in parallel on a single GPU. The representation states are sampled from a known distribution with learned parameters like Gaussian (Hafner et al., 2020) or Categorical (Hafner et al., 2021). The concatenation of outputs from the recurrent and image representation models gives the compact model state ($s_t = [h_t, z_t]$). The posterior model state is further used to reconstruct the original image $\hat{x}_t$, predict the reward $\hat{r}_t$, and discount factor $\hat{\gamma}_t$. The discount factor helps to predict the probability that an episode will end. The components of the world model are as follows:

$$
\begin{array}{lll}
\text{Recurrent model:} & h_t = f_\phi(h_{t-1}, z_{t-1}, a_{t-1}) \\
\text{Representation model:} & z_t \sim q_\phi(z_t \mid h_t, x_t) \\
\text{Transition predictor:} & \hat{z}_t \sim p_\phi(\hat{z}_t \mid h_t) \\
\text{Image predictor:} & \hat{x}_t \sim p_\phi(\hat{x}_t \mid h_t, z_t) \\
\text{Reward predictor:} & \hat{r}_t \sim p_\phi(\hat{r}_t \mid h_t, z_t) \\
\text{Discount predictor:} & \hat{\gamma}_t \sim p_\phi(\hat{\gamma}_t \mid h_t, z_t).
\end{array}
\tag{1}
$$

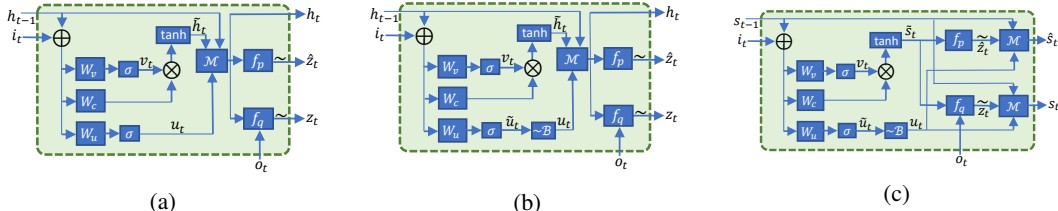

Figure 2: Architectures of (a) Recurrent State-Space Model (RSSM), (b) Variational Sparse Gating (VSG), and (c) Simple Variational Sparse Gating (SVSG), respectively. $\sigma$ and $\tanh$ denote the sigmoid and tanh non-linear activations, respectively. $W_*$ and $b_*$ are the corresponding weights and biases. $\sim$, $\oplus$ and $\otimes$ denote sampling, vector concatenation, and element-wise multiplication, respectively. $\mathcal{M}$ computes $x_t = u_t \tilde{x}_t + (1 - u_t)x_{t-1}$, where $x_t = h_t$ is used for RSSM and VSG, and $x_t = s_t$ is used for SVSG. $\mathcal{B}$ denotes Bernoulli distribution. $f_p$ and $f_q$ denote the prior and posterior distributions with learned parameters, respectively (See Appendix I for more details).

**Neural Networks**: The representation model outputs the posterior image representation state $z_t$ conditioned on the image encoding $x_t$ and recurrent state $h_t$. The transition predictor provides the prior image representation state $\hat{z}_t$. The image encoding $o_t$ is obtained by passing the image $x_t$ through CNN (LeCun et al., 1989) and Multi-layer Perceptron (MLP) layers. In VSG, we propose to modify the Gated Recurrent Unit (GRU) used in RSSM to sparsely update the recurrent state at each step. The model state $s_t$, which is a concatenation of recurrent and image representation states is passed through several layers of MLP to predict the discount and reward, and transposed CNN layers are used to reconstruct the image. The Exponential Linear Unit (ELU) activation is used for training all the components of the world model (Clevert et al., 2015).

**Sparse Gating**: In light of training RNNs to capture long-term dependencies, different ways of applying sparse updates have been investigated (Campos et al., 2017; Neil et al., 2016; Goyal et al., 2019), enabling a subset of state dimensions to be constant during the update. They were found to alleviate the vanishing gradient problem by effectively reducing the number of sequential operations (Campos et al., 2017). Discrete gates may also improve long-term memory by avoiding the gradual change of state values introduced by repeated multiplication with continuous gate values in standard recurrent architectures. Previous works on sparsely updating hidden states (Campos et al., 2017; Neil et al., 2016) use a separate layer applied over the outputs of RNN, and do not modify the RNN in itself. However, in this work, we modify the update gate in GRU (Cho et al., 2014) to take binary values by sampling from a Bernoulli distribution (Fig. 2 [b] shows the architecture).

The input $i_t$ to the recurrent model contains information about the action and is obtained by concatenating the previous image representation state $z_{t-1}$ and action $a_t$ followed by passing them through a MLP layer. Similar to GRU (Cho et al., 2014), there is a reset and update gate. The reset gate $v_t$ decides the extent of information flow from the previous recurrent state and inputs, and the update gate $u_t$ tells which parts of the recurrent state will be updated. The update gate takes only binary values, selecting whether the value will be updated or copied from previous time step. Binary values are obtained by sampling from a Bernoulli distribution where the probability of sampling is obtained using the previous recurrent state $h_{t-1}$ and input $i_t$. Straight-through estimators (Bengio et al., 2013) were used for propagating gradients backwards for training. The update equations are:

$$v_t = \sigma(W_v^T[h_{t-1}, i_t] + b_v)$$
$$\tilde{u}_t = \sigma(W_u^T[h_{t-1}, i_t] + b_u)$$
$$\tilde{h}_t = \tanh(v_t * (W_c^T[h_{t-1}, i_t] + b_c)) \qquad (2)$$
$$u_t \sim \text{Bernoulli}(\tilde{u}_t)$$
$$h_t = u_t \odot \tilde{h}_t + (1 - u_t) \odot h_{t-1},$$

where $\odot$ denotes element-wise multiplication, $\sigma$ and $\tanh$ are the sigmoid and hyperbolic tangent activation function, and $W_*$ and $b_*$ denotes the weights and biases, respectively. To control the sparsity of updates, we have used KL divergence between probability of sampling the update gate $\tilde{u}_t$ and a fixed prior probability $\kappa$, where $\kappa$ is a tunable hyperparameter.

**Loss function**: The predictors for image and reward produces Gaussian distributions with unit variance, whereas the discount predictor predicts a Bernoulli likelihood. The image representation

states are sampled from a Gaussian (Hafner et al., 2020) or a Categorical (Hafner et al., 2021) distribution which are trained to maximize the likelihood of targets. In addition, there is a KL Divergence term between the prior and posterior distributions and similar to DreamerV2 (Hafner et al., 2021), we have also used KL balancing with a factor of 0.8. We have also added a sparsity loss to regularize the number of updates in hidden state at each step. All the components of the world model are optimized jointly using the loss function given by:

$$\mathcal{L}(\phi) \doteq \mathrm{E}_{q_\phi(z_{1:T} \mid a_{1:T}, x_{1:T})} \Big[ \sum_{t=1}^{T} \underbrace{-\ln p_\phi(x_t \mid h_t, z_t)}_{\text{image log loss}} \underbrace{-\ln p_\phi(r_t \mid h_t, z_t)}_{\text{reward log loss}}$$

$$\underbrace{-\ln p_\phi(\gamma_t \mid h_t, z_t)}_{\text{discount log loss}} \underbrace{+\beta \, \mathrm{KL}\big[q_\phi(z_t \mid h_t, x_t) \,\big\|\, p_\phi(z_t \mid h_t)\big]}_{\text{KL loss}} \underbrace{+\alpha \, \mathrm{KL}\big[\tilde{u}_t \,\big\|\, \kappa\big]}_{\text{sparsity loss}} \Big], \tag{3}$$

where $\beta$ and $\alpha$ are the scale for KL losses of the latent codes and the sparse update gates, respectively.

## 3   Simple Variational Sparse Gating

Stochastic State-Space Model (SSM) were proposed in PLaNet (Hafner et al., 2019), where it was discussed that it is not trivial to achieve competitive results without the deterministic recurrent path. Having a deterministic component was motivated to allow the transition model to retain information for multiple time steps as the stochastic component induces variance (Hafner et al., 2019). In this work, we show that having a purely stochastic component achieves comparable performance with DreamerV2 while significantly outperforming SSMs (refer to Appendix H for more details). We introduce a simplified version of VSG, called Simple Variational Sparse Gating (SVSG) where the world model has a model state with single path to preserve information over multiple steps and also account for partial observability in future states (Fig. 2 [c] presents the SVSG architecture).

**Model Components**: In SVSG, there is no recurrent model and the posterior state $s_t$ is obtained using the representation model by conditioning on the previous state $s_{t-1}$, input image $x_t$ and the action $a_t$. Similar to VSG, there is a transition predictor that returns the prior state $\hat{s}_t$ which does not use the current image observation to imagine trajectories in the latent space. Both the modules sparsely update the model state at each step using the stochastic gating mechanism proposed in VSG. We have used a Gaussian distribution for the stochastic state with a learnable mean vector and a learnable diagonal covariance matrix. Similar to VSG, the posterior state is used to reconstruct the image, and predict the reward and discount factor. The components of world model in SVSG are:

$$
\begin{aligned}
\text{Representation model:} \quad & s_t \sim q_\phi(s_t \mid s_{t-1}, x_t, a_t) \\
\text{Transition predictor:} \quad & \hat{s}_t \sim p_\phi(\hat{s}_t \mid s_{t-1}, a_t) \\
\text{Image predictor:} \quad & \hat{x}_t \sim p_\phi(\hat{x}_t \mid s_t) \\
\text{Reward predictor:} \quad & \hat{r}_t \sim p_\phi(\hat{r}_t \mid s_t) \\
\text{Discount predictor:} \quad & \hat{\gamma}_t \sim p_\phi(\hat{\gamma}_t \mid s_t).
\end{aligned}
\tag{4}
$$

The representation model $q_\phi$ and transition predictor $p_\phi$ are modified to output the posterior $s_t$ and prior $\hat{s}_t$ states, respectively. The reset gate $v_t$ and the update gate $\tilde{u}_t$ is calculated using the previous state $s_{t-1}$ and input $i_t$ which has the information about the action $a_t$. The candidate state $\tilde{s}_t$ at each step is obtained using input $i_t$, reset gate $v_t$ and previous state $s_{t-1}$. Similar to VSG, the update gate $u_t$ is sampled from a Bernoulli distribution to sparsely update the latent states at each step, given by:

$$
\begin{aligned}
v_t &= \sigma(W_v^T[s_{t-1}, i_t] + b_v) \\
\tilde{u}_t &= \sigma(W_u^T[s_{t-1}, i_t] + b_u) \\
\tilde{s}_t &= \tanh(v_t \odot (W_c^T[s_{t-1}, i_t] + b_c)) \\
u_t &\sim \text{Bernoulli}(\tilde{u}_t),
\end{aligned}
\tag{5}
$$

where $\odot$ denotes the element-wise multiplication, $\sigma$ and $\tanh$ are the sigmoid and hyperbolic tangent activation functions, and $W_*$ and $b_*$ denote the weights and biases, respectively.

The candidate state $\tilde{s}_t$ is feeded through MLP layers to get the prior and posterior distributions. The image encoding $x_t$ was used to get posterior distribution, whereas the prior distribution was predicted without it. The prior $\hat{z}_t$ and posterior $z_t$ candidate states are sampled from these distributions, where

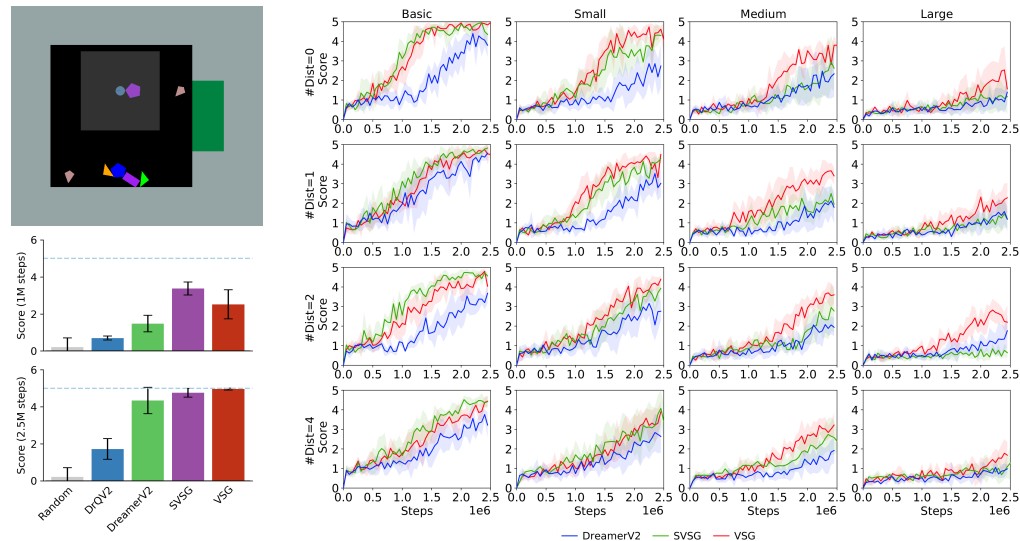

Figure 3: a) (Top Left) Full arena of the BringBackShapes (BBS) where the gray region around agent shows the partial view received by it. The circular agent is located in the center of partial view and is of teal blue color. The task is to push objects in the green goal region on the right side of arena. b) (Bottom left) Scores obtained on BBS with Basic size and no distractors at 1M and 2.5M steps. c) (Right) Performance (results over 5 seeds are reported) at different sizes of arena and number of distractor objects (#Dist). VSG and SVSG outperforms DreamerV2 significantly in most scenarios.

the update gate sparsely modifies the previous latent state and outputs the prior $\hat{s}_t$ and posterior $s_t$ model states at each step, respectively. The update equations are given by:

$$
\begin{aligned}
\hat{z}_t &\sim f_{\mathrm{p}}(\tilde{s}_t) \\
z_t &\sim f_{\mathrm{q}}(\tilde{s}_t, x_t) \\
\hat{s}_t &= u_t \odot \hat{z}_t + (1 - u_t) \odot s_{t-1} \\
s_t &= u_t \odot z_t + (1 - u_t) \odot s_{t-1},
\end{aligned}
\tag{6}
$$

where $f_{\mathrm{p}}$ and $f_{\mathrm{q}}$ denotes functions that output a distribution with learnable parameters for prior and posterior, respectively. For SVSG, Categorical latents (Hafner et al., 2021) were not performing well on our tasks. We attribute this to the fact that samples from a categorical distribution are binary vectors and it is difficult to accurately reconstruct with such sparse latent representations. Lastly, we observed that sparse gating mechanism introduced in VSG was important for convergence of SVSG.

**Loss function**: We have used the same loss function as described in Sec. 2 and policy is similar to used in VSG (described in Appendix B). For training the SVSG model, we replace the KL loss term between prior and posterior distributions in Eq. 3 with a masked KL loss that penalizes the state dimensions that were updated in the corresponding time step, i.e. those for which the corresponding element in $u_t$ is equal to 1. We found this to be necessary, since the original, unmasked KL loss did not yield good performance, presumably due to its effect on state dimensions that were not updated.

## 4 Experiments

### 4.1 BringBackShapes

**Environment**: In this work, we developed the BringBackShapes (BBS) environment to test the ability of agents to solve tasks in partially-observable and stochastic scenarios (see Fig. 3 [a]). The task is to push the objects within the arena into a pre-specified goal area. Moreover, rewards are sparse and is +1 for successfully pushing an object into the goal. At each time step, the agent only receives an obfuscated view of the arena centered around its current position. This requires agents to efficiently explore the arena to find new objects as well as remember states of previously observed objects. The objects can collide with each other and the walls, which further requires the agent to account for these events while updating its state. Stochasticity was introduced in the environment by

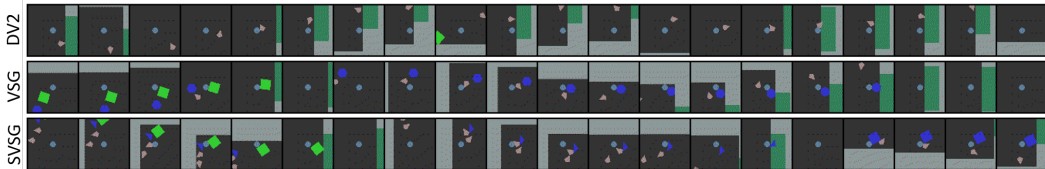

Figure 4: Learned behaviors of DreamerV2 (DV2), VSG and SVSG agents on BringBackShapes on Basic size and with 2 distractor objects at different steps. DreamerV2 fails to capture that distractors (whitish cones) are noisy objects and tries to push them towards the goal, whereas VSG and SVSG learn to avoid the noisy objects and carefully maneuvers the right objects towards the goal.

| | First-Visit Time | Episode Length | Objects Not Visited (%) | Visited Objects Not Scored (%) | First-Visit Time | Episode Length | Objects Not Visited (%) | Visited Objects Not Scored (%) |
|---|---|---|---|---|---|---|---|---|
| | Basic, #Distractors=0 | | | | Medium, #Distractors=0 | | | |
| DV2 | 500.25 | 2503.55 | 4.6 | 14.67 | 1800.79 | 2994.91 | 50.2 | 33.68 |
| VSG | **276.20** | **1881.80** | **0.08** | **1.38** | **1360.32** | **2953.21** | 32.6 | **24.98** |
| SVSG | 365.08 | 2196.37 | 2.00 | 5.15 | 1375.99 | 2964.85 | **30.00** | 38.53 |
| | Basic, #Distractors=2 | | | | Medium, #Distractors=2 | | | |
| DV2 | 593.53 | 2908.73 | 6.40 | 35.90 | 1669.42 | 2998.51 | 40.80 | 45.48 |
| VSG | 330.74 | 2482.40 | **0.4** | 9.67 | **1051.29** | **2944.34** | **16.4** | **32.12** |
| SVSG | **313.95** | **2292.00** | 0.6 | **7.61** | 1484.09 | 2988.79 | 31.80 | 51.42 |

Table 1: Average values of the first-visit time, episode length, % of objects not visited, and % of objects visited but not scored within an episode for trained agents on Basic and Medium environments, and with 0 and 2 distractor objects respectively. Metrics were calculated for 50 episodes for 5 seeds. VSG and SVSG significantly outperformed DreamerV2 on most statistics.

using random distractor objects. The distractor objects follow Brownian motion in any direction and can't be pushed into the goal area. Additionally, they add noise to the reward signal as they might push objects into the goal, causing a reward that is not or only partially related to the agent's behavior. Due to partial-observability, such instances might not be visible to the agent, making the task even more challenging. Refer to Appendix A for further description of the environment.

**Experimental Setup**: The BringBackShapes (BBS) environment returns high dimensional images of shape $64 \times 64 \times 3$ as observation. Action is a 2-dimensional continuous vector with acceleration and direction as components. Episodes last for 3000 environment steps and an action repeat (Mnih et al., 2016) of 4 was used. Baseline agents include DreamerV2 (Hafner et al., 2021) and DrQ-v2 (Yarats et al., 2022). In Appendix C, we mention the hyperparameters for the proposed methods- VSG and SVSG. The model was implemented using Tensorflow Probabability (Dillon et al., 2017) and trained on a single NVIDIA V100 GPU with 16GB memory. Training time for DreamerV2, VSG and SVSG methods on the BBS environment for 2.5M environment steps are around 12, 11 and 10.5 hours, respectively. Lastly, results are reported across 5 seeds. [1]

**Quantitative Results**: Fig. 3 [b] compares the proposed methods VSG and SVSG with the leading RL agents- DreamerV2 (Hafner et al., 2021) and DrQ-V2 (Yarats et al., 2022). The score indicates how many objects on average were scored within a episode. As discussed in Section 3, we trained SVSG with Gaussian latents only. Upon evaluation at 2.5M timesteps, DreamerV2 achieves competitive scores when compared to the proposed methods. Whereas at 1M steps, DreamerV2 does not perform as well as VSG, which has mean score of **4.9**. This shows that learning with sparsity priors helps improve the sample efficiency. Furthermore, performance of SVSG is better than DreamerV2 but similar to VSG, demonstrating that a purely stochastic model can achieve similar performance.

**Varying Partial-Observability and Stochasticy**: We also study the effect of partial observability and stochasticity. For partial-observability, we increased the size of the arena while reducing the portion visible to the agent. We consider 4 configurations of the arena- Basic, Small, Medium and Large. For stochasticity, we increase the number of distractor objects using values 0, 1, 2, and 4. Fig. 3 [c] presents the plots of models trained at 2.5M steps at different sizes of the arena and number

---

[1]Code is available at: https://github.com/arnavkj1995/VSG.

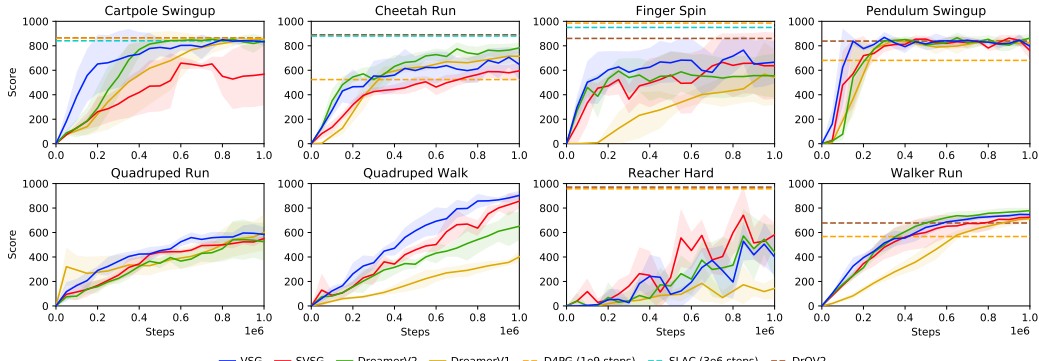

Figure 5: Comparison of VSG and SVSG with DreamerV1 (Hafner et al., 2020) and DreamerV2 (Hafner et al., 2021) on the DeepMind Control Suite. VSG converges faster on many tasks as demonstrated by evaluation curves. Even with a single stochastic path, SVSG achieves performance competitive to the models that use a combination of multiple paths.

of distractor objects. It can be observed that increasing the size of arena makes it harder to score objects. VSG was found to outperform DreamerV2 across all arena sizes. However, SVSG did not perform well on larger arena sizes. Adding noisy distractor objects led to drop in final performance of all models. But VSG and SVSG still outperformed DreamerV2, indicating that the sparsity prior helps in ignoring the noisy objects in the arena while solving the task.

**Ablation Studies**: In this work, we also report statistics to describe the behavior of learned agents. First-visit time is the number of episode steps taken to visit an object (when object is completely visible in the agent's view) and is calculated by averaging the first-visit time of each object in the arena. Lower first-visit time indicates that an agent is able to quickly discover all the objects in the arena. Another metric is Episode Length which denotes the number of steps taken by the agent to complete the task. The maximum of these scores was set to 3000. We also report the percentage of objects that were visited within an episode which represents the ability of agents to explore all parts of the arena to find novel objects. Lastly, we report the percentage of visited objects that were not scored which indicates us that the agent might not be remembering positions of objects seen previously. Thus, agents might have to explore the arena again to find them leading to an increase in time taken to finish the task. Table 1 presents the results on different settings of the environment and it can be observed that VSG and SVSG outperform DreamerV2 significantly.

**Qualitative Results**: We also observed the maneuvers taken by the agents to push the objects to the goal. The DreamerV2 agent was able to recognize objects and go behind them to push, but did not follow a smooth trajectory and was spending more time around an object to push it in the goal. However, our methods showed smoother trajectories and were more efficient at pushing objects successfully in the goal area. Additionally, our methods learned to avoid noisy distractor objects whereas the DreamerV2 agent was colliding with them and was trying to push them to the goal (see Fig. 4 and supplementary material for more videos).

**Effect of Sparse Gating**: We conducted an experiment where the learned world model was given the first 15 frames and 5 different rollouts were generated in the latent space for the next 35 frames. The sequence of actions is kept fixed across rollouts. The aim was to observe if the sparse gating mechanism is helping the model to retain information for longer time steps and the imagined trajectories are consistent. It was observed in Appendix J that learned world model in VSG and SVSG remembers the color and location of objects, and is also cognizant about the goal location and walls. Furthermore, unrolled trajectories from the world model of DreamerV2 showed distortion in the shapes, and in some instances modifies the color of the objects.

### 4.2 DeepMind Control Suite

**Experimental Setup**: The proposed method is evaluated on a few tasks from DeepMind Control Suite (Tassa et al., 2018). Observations for the agents are high dimensional images of shape $64 \times 64 \times 3$, actions range between 1 to 12 dimensions, and episodes last for 1000 steps. An action repeat (Mnih et al., 2016) of 2 was used. The model was implemented using Tensorflow Probabability (Dillon

et al., 2017) and trained on a single NVIDIA V100 GPU with 16GB memory in less than 6 hours. The agents were trained for 1M environment steps. Baselines include DreamerV1 (Hafner et al., 2020), DreamerV2 (Hafner et al., 2021), DrQ-v2 (Yarats et al., 2022), D4PG (Barth-Maron et al., 2018), and A3C (Mnih et al., 2016). Except A3C, all baselines learn policies from high dimensional pixel inputs. DreamerV2 was trained using the implementation provided by the authors. For other baselines, the metrics provided by the respective authors were used for comparison. Lastly, returns averaged across 5 seeds were reported.

**Results**: Figure 5 presents comparison of VSG and SVSG with the baseline agents (Refer to Appendix D for comparison on more tasks). It can be observed that on most tasks, VSG performs comparable to or better than DreamerV2 (Hafner et al., 2021). Notably, VSG significantly outperforms DreamerV2 on Quadruped and Finger-Spin tasks. Furthermore, SVSG with a purely stochastic component has similar performance to DreamerV2, outperforming on Finger Spin and Quadruped tasks and performing worse on Cartpole Swingup and Cheetah Run tasks. In addition, SVSG significantly outperforms DreamerV1 on many tasks which also used Gaussian latents and RSSM with multiple paths. Lastly, we also present the importance of sparsity loss in VSG (See Appendix I.3), and of KL Masking and Sparsity loss in SVSG (see Appendix E.2).

**Ablation Studies**: In BBS, we added noise in the environment by having distractor objects. We also experimented with other forms of noise where natural videos are used in the background for DeepMind Control tasks (Zhang et al., 2021; Nguyen et al., 2021b). Since reconstruction-free model based RL (Deng et al., 2021; Nguyen et al., 2021a) perform better than reconstruction based agents (Hafner et al., 2021) in such scenarios, we updated the RSSM block in DreamerPro (Deng et al., 2021) with VSG and call it VSGPro, and trained it on DMC with natural background (Refer to Appendix F). We also experimented with VSG in discrete control tasks from the Atari benchmark in Appendix G, where VSG performed better on task with changing viewpoints.

## 5 Related Work

**Latent Dynamics Models**: Latent dynamics models (Bourlard and Morgan, 2012; Kalman, 1960; Bengio et al., 1999) operate directly on sequences predicted in the latent space rather than autoregressively feeding back the generated frames back to the model. Recent advancements in deep learning have allowed learning expressive latent dynamics models using stochastic backpropagation (Kingma and Welling, 2013; Chung et al., 2015; Krishnan et al., 2015; Karl et al., 2016). Recurrent State-Space Model (RSSM) (Hafner et al., 2019) comprises of stochastic and deterministic components. VideoFlow (Kumar et al., 2019) predicted future values of the latent state by normalizing flows for robotic object interactions. Hierarchical latent models for video prediction were proposed in CWVAEs (Saxena et al., 2021) with levels ticking at different intervals in time. (Franceschi et al., 2020) and (Donà et al., 2021) disentangled dynamic and static factors where 2-5 initial observations was used to estimate the static component.

**RL for Visual Control**: Deep Reinforcement Learning (DRL) methods fall into one of two categories: 1) *Model-Based* — where an explicit model of the environment and its dynamics are learned (Ha and Schmidhuber, 2018; Hafner et al., 2019; 2020; 2021; Zhang et al., 2019; Kaiser et al., 2019), and 2) *Model-Free* — where a policy is learned directly from the raw observations (Srinivas et al., 2020; Kostrikov et al., 2020b; Lillicrap et al., 2015; Yarats et al., 2022; Schwarzer et al., 2021; Mondal et al., 2022). Deep Deterministic Policy Gradient (DDPG) (Lillicrap et al., 2015) combined actor-critic with insights from DQNs (Mnih et al., 2015) to learn agents for continuous action spaces. TD3 (Fujimoto et al., 2018) builds upon the DDPG algorithm and addresses the problem of overestimation bias in the value function. CURL (Srinivas et al., 2020) uses contrastive losses to learn discriminative representations. DrQ (Kostrikov et al., 2020a) and DrQ-v2 (Yarats et al., 2022) employed data augmentation techniques and do not use auxiliary losses or pre-training. RSSM was introduced in PlaNet (Hafner et al., 2019) and was employed for online planning in the latent space. DreamerV1 (Hafner et al., 2020) and DreamerV2 (Hafner et al., 2021) achieved state-of-the-art results on DMC (Tassa et al., 2018) and Atari (Bellemare et al., 2013), respectively. SimPLe (Kaiser et al., 2019) trains a PPO (Schulman et al., 2017) agent on the learned video generation model in pixel space. SOLAR (Zhang et al., 2019) solved robotics tasks via guided policy search.

**Sparsity in RNN**: Neural networks have widely adopted sparsity to reduce the memory footprint of weights and activations (LeCun et al., 1990; Chen et al., 2015; Han et al., 2015). Several works have explored sparsity in RNNs. Campos et al. (2017) introduced a mechanism in RNNs that learns

to skip state updates, effectively reducing the number of sequential operations on the latent state, thereby alleviating the problem of vanishing gradients in training on long sequences. Goyal et al. (2019) presented Recurrent Independent Mechanism (RIM), an architecture that consists of separate recurrent modules which are sparsely updated using a learned attention mechanism. In contrast to RIM, the number of updated state variables in VSG algorithm is not fixed.

## 6  Discussion

In this work, we introduce VSG and SVSG, two latent dynamics models leveraging sparse state updates. The sparse update prior was found to facilitate more efficient behaviors in tasks requiring long-horizon planning. Furthermore, SVSG is a purely stochastic model with a single component in the model state. We show that VSG and SVSG can outperform leading agents on the proposed BringBackShapes task, a challenging partially-observable and stochastic environment. BBS allows for controlling different factors of variation like stochasticity and partial-observability. Experiments conducted on various variations in BBS demonstrate that the proposed agents are more robust to noise in the environment and can better retain information of seen objects. Some limitations and potential research directions for future research are as follows:

- In the current implementation of VSG, the latent space does not exhibit disentanglement which could be an interesting direction for future research. Gating mechanisms in VSG can also be combined with other recurrent architectures like RIM (Goyal et al., 2019).

- In this work, BBS was explored with only 2 factors of variation: partial-observability and stochasticity. More controllable factors like the nature of entities (shape, size, color of objects), underlying physics (mass, friction, elasticity), or procedural background generation can be introduced to further study generalization capabilities of RL agents.

- SVSG being a purely stochastic model can further be used to estimate state uncertainty by marginalizing over multiple samples paths to efficiently explore in an unknown environment.

- Evaluation on first-person view 3D games like tasks in DMLab (Beattie et al., 2016) would be interesting. Furthermore, a 3D version of the BBS environment with the viewpoints changing with rotation of agent and the underlying physics will make the task more challenging.

- We have used small latent dimensions and it would be interesting to train such models with larger architectures and on more complex tasks. Scaling the current architecture would also require optimizing the implementation to make them computationally feasible.

- Categorical latents outperformed Gaussian latents as the stochastic states of RSSM (Hafner et al., 2021), especially for discrete control tasks. However, SVSG was not found to work well with Categorical latents and we believe that sampled sparse states are hard to optimize.

- Model-based RL for visual control is still in early stages. However, a major challenge with deploying such models in the real world is safety especially during exploration. This would require an accurate world model that allows learning policies with stringent safety constraints that avoid mistakes when deployed in the real world. Such algorithms will rely on models that are robust when transferred from simulation to the real world.

## Acknowledgements

The authors would like to thank David Meger, Lucas Lehnert and Ankesh Anand for their valuable feedback and discussions. The text also benefited from discussions with Abhinav Agarwalla, Rupali Bhati and Vineet Jain. The authors are also grateful to CIFAR for funding and the Digital Research Alliance of Canada for computing resources.

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
