# OpenReview forum: "Learning Robust Dynamics through Variational Sparse Gating"
_NeurIPS.cc/2022/Conference — NeurIPS 2022 Accept_

### Official Review · Reviewer_Dxye · 2022-07-10

**Rating:** 7
**Confidence:** 3
**Soundness:** 3 good
**Presentation:** 3 good
**Contribution:** 2 fair

**Summary:**

This paper proposes two new latent dynamics models (VSG and SVSG), which aims to incorporate a certain inductive bias: in some real world problems, the changes are very sparse, so the latent can be updated sparsely to help long term memory. They test the proposed algorithm on a new environment BBS, which shows that VSG and SVSG outperforms existing methods, and on DMC benchmark, which shows that VSG and SVSG ad comparable with existing methods.

**Questions:**

See above

**Strengths And Weaknesses:**

Strengths:
- This is a very interesting paper to incorporate the inductive bias to existing latent dynamics model RSSM and SSM, and the results show that the proposed methods seem to be effective on specific problems.
- It is also every interesting to see that after the modification, the SVSG can outperform SSM and comparable with RSSM/VSG.

Weakness:

I agree that this is a very interesting work, but I think it will be better if the analysis can be more sufficient. For instance, by *randomly* selecting some latents $u_t$ to be zero in GRU, the VSG can achieve better performance on long-memory cases, which is good. However, this may mismatch the motivation “the real world changes are sparse” since the selection is purely stochastic. And this is even more significant for SVSG. I think it is important to understand why the proposed methods may work.
Can I think of this as a regularization trick, which ask for the encoder to encode the information duplicately for each dimension, since each of them might be zeroed? Is it possible for you to show an experiment that shows the effects of the randomness? For instance, given a sequence, and several different samples from Bernoulli distributions, to see if the next state can decode the same image?

---

> ### Author Response · Authors · 2022-08-02
> **Response to Reviewer Dxye**
>
> We would like to thank you for your valuable suggestions and address the weakness/questions below:
>
> > Can I think of this as a regularization trick, which asks for the encoder to encode the information duplicately for each dimension, since each of them might be zeroed?
>
> We believe that it is possible that the model learns to associate multiple hidden units for an object. Therefore, we feel it would be an interesting direction for future work to incorporate inductive biases from architectures that allow us to interpret the hidden state like Recurrent Independent Mechanisms (RIMs) (https://arxiv.org/abs/1909.10893)  or Slot Attention (https://arxiv.org/abs/2006.15055). Lastly, having duplicity for each object is also possible in methods like RIMs, where it is assumed that each module has its own hidden state to represent an object and is composed of multiple units.
>
> > However, this may mismatch the motivation “the real world changes are sparse” since the selection is purely stochastic.
>
> We agree with the point that the selection is purely stochastic, however the model might learn to have high probabilities for parts of the hidden state that is to be updated. As discussed in limitations, the current hidden state does not exhibit disentanglement. Also, adding such sparse update mechanisms to RIMs should help study this direction as RIMs have been found to show disentanglement in the latent space.
>
> > Is it possible for you to show an experiment that shows the effects of randomness?
>
> We did an experiment where we give the learned world model the first 15 frames and ask it to imagine 5 different rollouts in the latent space for the next 35 frames. The sequence of actions is kept fixed across rollouts. It can be observed from the GIFs of VSG and SVSG that the model is able to remember the color and location of objects, and is also cognizant about the goal location and walls. Furthermore, for GIFs of DreamerV2, it can be observed that there is a distortion in the shapes and the model also modifies the color of the objects towards the end. This further shows that the proposed mechanism is helping the model to retain information for longer time steps. Please refer to Appendix J for more details.
>
> We hope we addressed most of the questions and hope you consider updating your score.

---

> > ### Author Response · Authors · 2022-08-09
> > **Follow-up on Rebuttal**
> >
> > Dear Reviewer Dxye,
> >
> > Following your suggestions, we have added an ablation to show the effect of randomness induced by the sampling of gates. In addition, we have discussed information duplicity in the embeddings.
> >
> > Kindly consider reassessing our work taking into account the modifications. Please let us know if your concerns have been addressed and we hope you increase the score.
> >
> > Thank you for your time and feedback,
> >
> > The Authors

---

### Official Review · Reviewer_TPk3 · 2022-07-11

**Rating:** 5
**Confidence:** 4
**Soundness:** 3 good
**Presentation:** 3 good
**Contribution:** 2 fair

**Summary:**

This paper proposes two recurrent models for model-based RL: Variational Sparse Gating (VSG), improved from DreamerV2 with binary gate, and Simple Variational Sparse Gating (SVSG) which has purely stochastic states. It also introduces a new partially-observable environment, BringBackShapes(BBS), to evaluate agents under POMDP setting. The experimental results show that the proposed VSG outperforms DreamerV2 in continuous control tasks including BBS and DeepMind Control Suite.

**Questions:**

1. Setting $u_t$ as binary values seems make the model not recurrent anymore. The model parameters $W_v, b_c, W_c$ and $b_c$ only get update when $u_t = 1$.
2. I am afraid that $u_t=0$ would be a big issue when planning in world model (as Figure 1(b)). If $ u_t=0 $, then $ h_t = h_{t-1} $ due to Equation (2).
The transition predictor $ p_{\phi}(z_t|h_t) $ that predicts prior state $ \hat{z}_t $
depends only on $ h_t $.

  In the case of $h_t = h_{t-1} $, the predicted state $ \hat{ z }_t $

  would be similar to $ \hat{z}_{t-1} $ at time step $t-1$.
Consequently, the actor conditioned on $ \hat{z}_t $ output the similar policy as time step $t-1$.

  If the distributions of $ p_{\phi}(z_t|h_{t-1}) $ and the policy have low variance, making $u_{k \geq t}=0$ almost surely, then the whole planning in world model takes input of the same hidden state $ h_{k \geq t} = h_{t-1} $ .

3. Does SVSG have the same losses as VSG ?
4. Why VSG and SVSG are called "variational"? It seems that there is nothing related to variational inference.
5. What setting of BBS is used in Figure 3b), i.e., the number of distractors and the size of arena?
6. In Table 1, what does first-visit time mean? Is it the time when visiting the first object, or the first time when visiting the last object (all objects)?
7. The experiments and related work lack of an important baseline method, SLAC[1], for continuous control tasks. SLAC solves POMDP by stochastic sequential models and also defines stochastic latent states.
8. DrQV2 is a strong baseline algorithm in continuous pixel control. Showing training curves of DrQV2 in figures would be good comparisons.
9. DeepMind Control Suite is POMDP but many algorithms assuming fully-observability solve DMC by stacking frames. It would be nice if you evaluate the methods on other public environments that are "truly" POMDPs, e.g., first-person view 3D games.

Other comments:
1. In line 85, $o_t$ is defined as state but in line 115 $s_t=[h_t,z_t]$ is also defined as state. I suppose the latter one should be correct for the definition of POMDP, since $o_t$ is the image encoding as in line 120.
1. SLAC is listed in the legend of Figure 5, but not shown in any curves or main text.

Reference

[1] Lee et al. Stochastic Latent Actor-Critic: Deep Reinforcement Learning with a Latent Variable Model. NeurIPS 2020


**Limitations:**

The effect of sparse gating needs to be further discussed and verified.

**Strengths And Weaknesses:**

## Strengths
1. VSG achieves impressive experimental results comparing to DreamerV2.
1. A new environment with various difficulty levels is designed for POMDP.

## Weakness
1. The significance of the proposed sparse gating needs to be clarified. In VSG, the sparse gating is the only improvement from DreamerV2, while the rest part including architecture and losses are the same as DreamerV2. The authors should discuss more about why sparse gating can improve performance.
For example, list some hypotheses to the reasons and verify them in experiments.
Besides, additional ablation studies about sparse gating may also help to demonstrate its effect.
Otherwise, without such discussion and verification, the technical contribution of VSG is considered as minor.
2. The motivation of SVSG is unclear. Previously Stochastic State-Space Model (SSM) that has stochastic state has been empirically proved to have worse results than RSSM that has deterministic state. In the experiments of this paper, SVSG also has worse performance than VSG. The authors should clarify the motivation of proposing a new model with stochastic state.

---

> ### Author Response · Authors · 2022-08-02
> **Response to Reviewer TPk3 (Part 1)**
>
> We would like to thank the reviewer for their thoughtful and insightful review, and address the weakness/questions below:
>
> > Why sparse gating can improve performance?
>
> Sparse updates in RNNs have aided them to retain memory for longer time steps. We believe that sparse updates introduced in VSG helps with long-term memory, and to verify this, we performed some analysis in Table 1 in the paper. We also tried adding sparsity by thresholding with prior probability ($\tilde{u}_t > \kappa$) and implemented SkipRNN (https://arxiv.org/pdf/1708.06834.pdf) , but they did not perform well.
>
> > Setting $u_t$ as binary values seems to make the model not recurrent anymore.
>
> Since $u_t$  is a vector of values, where each component denotes if a particular cell of hidden state will be updated at each step. We assume that $u_t$ is not a 0 vector and some components are 1, which means that $h_t \neq h_{t-1}$ as at least a single component in the hidden state will be updated. We further verified that at each step, some of the update gate values are always 1. Since the hidden states are not the same, the latent and policy outputs will be different for different time steps.
>
> > Does SVSG have the same losses as VSG?
>
> Yes, we have added this at line 198 in the paper.
>
> > Which setting of BBS is used in Fig 3b?
>
> The Basic version with 0 distractors is used to report the results. We have added this detail in the paper.
>
> > In Table 1, what does first-visit time mean?
>
> The first-visit time is calculated by obtaining the time step when an object was seen for the first time and taking the average across all objects in the episode. Visiting time of the first object depends a lot on the initialization of the episode. For example, if the agent sees the first object at the start then the first visit count is 0. The last object’s first-visit time can be used, but we believe that the Episode length can capture it as for solving the task, it is required to find all the objects. Thus, we took the average across all the objects.
>
> > Comparing with SLAC on BBS environment:
>
> Thank you for suggesting an important baseline. We ran SLAC on the basic version of BBS with 0 distractors and it was not found to work well achieving scores of 0.88  and 1 at 1M and 2.5M environment steps (across 5 seeds), respectively.
>
> > Training curves of DrQ-V2 in figures
>
> We agree that it would be interesting to add DrQ-v2 as baseline in the plots. However, we skipped it as DrQ-v2 was not doing very well in the Basic environment with 0 distractors and it is computationally complex to run on all the scenarios. We plan to add a separate curve for the Basic setting with 0 distractors and add curves of DrQ-v2 and SLAC to it.
>
> > Experiments on “first-person” view of games:
>
> In the 4th point in limitations (Section 6), we discussed about creating a 3D version of the BBS environment where the observation can be the agent's viewpoint. We tried a basic 3D version of BBS with agent's view as observation, but observed that all the agents were able to solve the task. We believe that in such a setting, agents were converging to simple policy which rotates to find the object and moves accordingly to push them towards the goal. We agree that it would be interesting to evaluate first-person view 3D games like tasks in DMLab. Furthermore, it would be interesting to add a maze-like 3D environment to BBS to make the task even more challenging. We will add them to the limitations and future works.
>
> > In line 85, $o_t$ is defined as state but in line 115 $s_t$ is also defined as state
>
> Thanks for pointing it out, we have updated the line 85 as $o_t$ should be observation.
>
> > SLAC is listed in the legend of Figure 5, but not shown in any curves or main text.
>
> Sorry about the mistake in Fig. 5. We did add the results for SLAC, but didn’t change the color of the line in Fig. 5. We have updated it in the main paper.

---

> > ### Author Response · Authors · 2022-08-02
> > **Response to Reviewer TPk3 (Part 2)**
> >
> > > Motivation of SVSG:
> >
> > In the context of this paper, we wanted to show that purely stochastic models can achieve competitive performance. We believe this is a significant contribution since the PLaNet paper showed that it is not trivial to achieve good performance without a deterministic path. The research on having fully stochastic latent space models is in very nascent stages and this work closes the gap by achieving results on par with leading algorithms. We believe having fully stochastic latent state-space models also paves the  way for interesting directions for future works like:
> >  - Obtaining state uncertainty estimates for guided exploration (similar to https://arxiv.org/abs/1902.07685).
> >  - Using Bisimulation metrics based methods for learning reconstruction-free World Models (as done in https://arxiv.org/abs/2006.10742).
> >  - Having a mixture of categorical latents would be interesting as VQ-VAE (https://arxiv.org/pdf/1711.00937.pdf) have been found to generate high quality images.
> >
> > > Additional ablation studies about sparse gating:
> >
> > We did an experiment where we give the learned world model the first 15 frames and ask it to imagine 5 different rollouts in the latent space for the next 35 frames. The sequence of actions is kept fixed across rollouts. It can be observed from the GIFs of VSG and SVSG that the model is able to remember the color and location of objects, and is also cognizant about the goal location and walls. Furthermore, for GIFs of DreamerV2, it can be observed that there is a distortion in the shapes and the model also modifies the color of the objects towards the end. This further shows that the proposed mechanism is helping the model to retain information for longer time steps. Please refer to Appendix J for more details.
> >
> > > Why “variational” in VSG and SVSG?
> >
> > In this work, we have used the term VSG to define the whole block as shown in Fig. 2b. Similar to RSSM, the block generates both prior and posterior latent states, hence we have used the term variational in the name. The sparse gating represents the proposed gating mechanism. Hence, we came up with the names- Variational Sparse Gating (VSG) and Simple Variational Sparse Gating (SVSG).
> >
> > We hope we addressed most of the questions and hope you consider updating your score.

---

> > > ### Author Response · Authors · 2022-08-09
> > > **Follow-up on Rebuttal**
> > >
> > > Dear Reviewer TPk3,
> > >
> > > Following your suggestions, we experimented with SLAC on BBS and it was not found to work well. In addition, we have provided an explanation for the term “variational” used in the context of our proposed methods VSG and SVSG. We also explained why setting $u_t$ will still yield a recurrent model except for the edge case where all elements are zero. We further added the description of the first-time visit count. We have fixed some issues in the revised manuscript (given the space limit) and will add other details later.
> > >
> > > Please consider reassessing our work. If you believe your concerns were addressed, we hope that you would be willing to increase your score.
> > >
> > > Thank you,
> > >
> > > The Authors

---

### Official Review · Reviewer_4uKx · 2022-07-11

**Rating:** 5
**Confidence:** 4
**Soundness:** 3 good
**Presentation:** 3 good
**Contribution:** 3 good

**Summary:**

This paper studies modelling the latent dynamics model by exploiting the idea that different elements of the latent space might exhibit varying degree of changes. The authors propose variational sparse gating (VSG), which is a modification of update gate of the GRU network (Cho, et al. 2014) in the recurrent state space model (RSSM; Hafner, et al. 2019) with a stochastic gating mechanism. The authors also propose Simple VSG (SVSG), which follows the similar stochastic gating updates as VSG, but contains only a purely stochastic latent state. VSG and SVSG are empirically evaluated on DeepMind Control Suite and a novel environment (BringBackShapes) to test their ability of solving stochastic tasks with partial-observability.

**Questions:**

The major concerns are listed in the previous section, some minor questions are listed below.

- What is the relative computational complexity between VSG and SVSG?
- Is the sparse reward setting necessary in BBS? A complicated (stochastic, non-Markovian) reward structure might be less trivial for standard agents without sparse gating (e.g., Dreamer v2) to acquire, according to the statements in the main text;
- The Private Eye evaluation (Figure 10) and the accompanying discussion is not so convincing given the large variance in the evaluation curves;

**Limitations:**

The authors have provided thoughtful discussion on the limitation and potential future areas for investigation of the VSG model, such as combination with other recurrent architectures, increasing the complexity of BBS environment, VSG dealing with categorical latents, etc.

I see no potential negative societal impact.

**Strengths And Weaknesses:**

Pros:
- The paper is well-written;
- The proposed new environment, BringBackShapes, is designed thoughtfully and sufficiently complicated to assess the ability of agents for solving tasks with stochastic transition and reward structures;

Cons:
- The motivation for SVSG is not clear, having a purely stochastic latent state does not show performance improvement in general, and no evidence/intuition of why this could be preferred over VSG (e.g., model complexity, stronger generalisability, etc.);
- There is inconsistency in the performance of VSG/SVSG with respect to varying level of stochasticity and partial-observability, e.g., one would expect monotonic increasing size of performance gap between VSG/SVSG and Dreamer V2 in the BBS environment, but none is observed (at least from the reported results in Figure 3c). Moreover, in the Atari evaluations (Figure 10), tasks such as Ms Pacman and Seaquest clearly exhibits strong stochasticity, but VSG fail to outperform Dreamer V2.

---

> ### Author Response · Authors · 2022-08-02
> **Response to Reviewer 4uKx**
>
> We thank you for your valuable and constructive feedback and address the questions/weaknesses below.
>
> > Computational complexity difference between VSG and SVSG-
>
> In Appendix C, we mentioned the total number of parameters for VSG and SVSG. The number of parameters are the same for DreamerV2 and VSG. Training time for DreamerV2, VSG and SVSG on the BBS environment for 2M environment steps are around 12, 11 and 10.5 hours on a NVIDIA V100 GPU with 16 GB memory, respectively.
>
> > Is the sparse reward setting necessary in BBS? A complicated (stochastic, non-Markovian) reward structure might be less trivial for standard agents without sparse gating (e.g., Dreamer v2) to acquire, according to the statements in the main text
>
> In this work, we focus on the sparse reward setting in the BBS environment. We can have dense rewards too, where the average distance of each object and goal is used. We tried under this dense reward setting, but all the agents were able to solve the task. BBS was designed to have sparse rewards, which makes the problem harder as the agent has to explore to get the first reward. Secondly, reward can occur due to an event that occurred earlier in time. For example, if an object was pushed earlier towards the goal, the reward will be visible when the object enters the goal. Thirdly, partial-observability makes it even harder because the agent’s view might not capture the movement of objects in the goal. Overall, solving tasks in BBS requires having better memory and makes the credit assignment more challenging.
>
> > Motivation of SVSG :
>
> In the context of this paper, we wanted to show that purely stochastic models can achieve competitive performance. We believe this is a significant contribution since the PLaNet paper showed that it is not trivial to achieve good performance without a deterministic path. The research on having fully stochastic latent space models is in very nascent stages and this work closes the gap by achieving results on par with leading algorithms. We believe having fully stochastic latent state-space models also paves the  way for interesting directions for future works like:
>  - Obtaining state uncertainty estimates for guided exploration (similar to https://arxiv.org/abs/1902.07685).
>  - Using Bisimulation metrics based methods for learning reconstruction-free World Models (as done in https://arxiv.org/abs/2006.10742).
>  - Having a mixture of categorical latents would be interesting as VQ-VAE (https://arxiv.org/pdf/1711.00937.pdf) have been found to generate high quality images.
>
>
> > There is inconsistency in the performance of VSG/SVSG with respect to varying level of stochasticity and partial-observability:
>
> With the increasing size of the arena, it gets harder to score in the environment, thus the maximum score achieved by the agent decreases with the increase in size. When comparing VSG and DreamerV2, we can see that the gap is nearly the same, but if we compare the percentage change in scores (since the scores achieved by agents reduces by increasing size of area), we can see an increasing trend.
> Moreover, increasing stochasticity (number of distractor objects) does not have drastic effect on the final scores, but as observed in GIFs (supplementary material and Fig 4), VSG/SVSG learns to avoid distractor objects whereas DreamerV2 agent tries to push the distractor in the goal area. This is even visible in environments with many distractor objects.
>
> > The Private Eye evaluation (Figure 10) and the accompanying discussion is not so convincing given the large variance in the evaluation curves.
>
> We agree with this point and ran VSG on Private Eye task for 3 more seeds. We observed that some of the seeds are not performing well which is leading to high variance. We have modified the discussion of results accordingly.
>
> We hope we addressed most of the questions and hope you consider updating your score.

---

> > ### Author Response · Authors · 2022-08-09
> > **Follow-up on Rebuttal**
> >
> > Dear Reviewer 4uKx,
> >
> > We would like to highlight that we have provided comparisons of computational complexity between algorithms, and the reasoning behind using sparse reward setting in BBS. In addition, we have elaborated on the motivation of SVSG and discussed the results. We will add these details in the revised version of the manuscript.
> >
> > We would like to kindly ask you to review the changes and assess whether the concerns are addressed. If so, we hope that you would be willing to increase your score.
> >
> > Thank you for your time,
> >
> > The Authors

---

> > > ### Comment · Reviewer_4uKx · 2022-08-09
> > > **Thanks for your replies**
> > >
> > > I wish to thank the authors for the replies to my questions. Most of the concerns are resolved for now but two remain.
> > >
> > > - Regarding the sparse-reward nature of BBS task.
> > >
> > > I appreciate the complexity of the BBS task due to its partial-observable and highly stochastic (even adversarial in some sense) nature. As the authors noted in their reply, experiments on dense reward settings of the BBS task lead to little improvement over baseline agents. Hence I wonder how does the VSG/SVSG agents perform on standard maze-like tasks with sparse rewards (e.g., minigrid). Another interesting point would be that have the authors examined the agent's ability for learning long-term contingencies (e.g., key-to-door tasks), this latter point is only for discussion and will not affect my assessment.
> > >
> > > - Regarding the motivation of SVSG.
> > >
> > > I can see how SVSG could be useful for computing intrinsic reward such as epistemic uncertainty (through posterior inference I assume) for exploration, however how this compares to ensemble-based methods, e.g., plan2explore (Sekar, et al. 2020), is questionable. My major concern is that SVSG does not provide better/more robust intrinsic reward for exploration than the simple ensemble-based methods, with the increased difficulty for training, hence I do not see strong motivation for the development of SVSG (which is a major component of the paper). Moreover, I fail to draw the connection between purely stochastic LSSMs with learning with bisimulation metrics (Zhang, et al. 2020).

---

> > > > ### Author Response · Authors · 2022-08-09
> > > > **Response to Reviewer 4uKx**
> > > >
> > > > Dear Reviewer 4uKx,
> > > >
> > > > > Comparison to ensemble-based methods, e.g., plan2explore:
> > > >
> > > > As discussed in Table 1 and Section 3.8 in APD (https://arxiv.org/abs/2009.01791), exploration using ensemble-based methods like Plan2Explore and using SVSG will fall under two different categories. Planning using methods like Plan2Explore will fall under model-uncertainty estimates where the goal is to discover the rules of the environment. Whereas planning using SVSG will fall under state uncertainty estimates where the goal is to disambiguate an unobserved environment. To the best of our knowledge, state uncertainty estimation has not been explored in the context of model-based RL for architectures like Dreamer and would be an interesting direction to explore.
> > > >
> > > > > Connection between purely stochastic LSSMs with learning with bisimulation metrics:
> > > >
> > > > Current implementations of DreamerV2 and VSG use a reconstruction loss in the pixel space. DreamerPro (https://arxiv.org/abs/2110.14565) introduced a reconstruction-free Model-based RL method. They outperform baselines on DMC with Natural background. However, they don't do better than DreamerV2 on Atari and DMControl tasks. Moreover, model-free algorithms like SPR (Schwarzer et al. 2021), DBC have relied on self-supervised learning for learning robust latent reconstructions. We believe that reconstruction-free model-based RL is an interesting avenue to explore in future works.
> > > >
> > > > DBC (Zhang et al. 2020) applied bisimulation metrics where a Wasserstein distance was used on outputs of the dynamics model (which returns a Gaussian distribution). Since SVSG only has a purely stochastic component, it would be interesting to replace the reconstruction based loss function with bisimulation loss to learn robust latent representations.
> > > >
> > > > > Experiments on Minigrid:
> > > >
> > > > We have not experimented with the key-to-door tasks. We have launched this experiment. We will add these results to the revised version once the model is trained. We thank you again for your constructive feedback. Please let us know if major concerns have been addressed and please consider re-evaluation if this is the case.

---

### Official Review · Reviewer_ANCN · 2022-07-12

**Rating:** 6
**Confidence:** 3
**Soundness:** 3 good
**Presentation:** 4 excellent
**Contribution:** 2 fair

**Summary:**

In summary, this paper provides a modification to the Gated Recurrent Unit (GRU) that *sparsely* updates the recurrent state by samplying from a Bernoulli distribution for the *update* term. Thus, rather than being a continuous value in GRU, this is now from $\{0, 1\}$. This is the Variational Sparse Gate (VSG).

The Simple Variational Sparse Gate (SVSG) further modifies VSG to be made purely stochastic (supposedly - see Questions), and used to evaluate claims from a previous paper that a deterministic recurrent path is required fro competitive results.

Experiments were carried out on a novel environment, and evironments from the DeepMind Control Suite. The novel environment specifically emphasizes the key aspects of this work, namely partial observability, and remembering past states.

**Questions:**

* Figure 2: in the caption, the description of $M$ does not match how it is used in Figure 2(c).
* line 54 and line162-164: mentions difference between a *purely stochastic* component, and one that contains some deterministic parts. I'm confused by what this means. From Figure 2, it looks like all units contain both deterministic and stochastic calculations. I'm probably missing something, could you say what that could be?
* line 263: Can you really say that, generally speaking, SVSG is comparable to DreamerV2? SVSG seems to perform worse in the Cartpole and Cheetah environments.


*General*
* What exactly is the role of the transition predictor?
* What is th meaning of *variational* in the name?
* How exactly do the components in Equations 1, 2, 4 fit together (in their separate cases)?
* How many seeds were used in Figure 3? Was it 3 (inferred from the checklist). If so, why is it different to the 5 seeds used in other figures?


**Limitations:**

I think limitations and future work are discussed well.

**Strengths And Weaknesses:**

*Strengths*
* Well written for the most part. Explanations and arguments are generally clear.
* Contributions are clearly stated, and the paper is structured will to argue these.
* A satisfying range of experiments was carried out, at least in the simpler case. The ablation studies were nice to see.
* Experiments were also carried out on more complex environments too.


*Weaknesses*
* Interpretations of the notation (Equation 1, 2, 4) could have been written more clearly.
* Lacks a baseline description of how GRU works (at least intuitively). This could help readers understand why the proposed changes are meaningful, and significant.

---

> ### Author Response · Authors · 2022-08-02
> **Response to Reviewer ANCN**
>
> We highly appreciate your constructive and valuable feedback to improve the paper.  We address the questions below:
> > Lacks a baseline description of how GRU works:
>
> We have added a basic description of RSSM with GRU as the recurrent model in Appendix I.
> > Figure 2: in the caption, the description of M does not match how it is used in Figure 2(c).
>
> Thank you for pointing this out, we have updated the description of Fig. 2 to use $h_t$ for VSG/RSSM, and $s_t$ for SVSG architecture. The role of M block is to update the state with candidate state and previous state. We have also made some edits to the diagram and added high resolution versions in Appendix I.
> > line 54 and line 162-164: mentions difference between a purely stochastic component, and one that contains some deterministic parts. I'm confused by what this means. From Figure 2, it looks like all units contain both deterministic and stochastic calculations.
>
> In this work, components denote output of the model at each step. In RSSM and VSG, the model state at each step ($s_t$/$\hat{s}_t$) are composed of 2 components at each step- the deterministic part ($h_t$) obtained from the recurrent model and stochastic part ($z_t$/$\hat{z}_t$) which is sampled from a learned probability distribution. Whereas in SVSG, there is only a single component which is sampled from a probability distribution with learned parameters.
> > line 263: Can you really say that, generally speaking, SVSG is comparable to DreamerV2? SVSG seems to perform worse in the Cartpole and Cheetah environments.
>
> SVSG does not perform as well as DreamerV2 on Cartpole and Cheetah environments, however it outperforms DreamerV2 on FingerSpin and Quadruped environments, and performs comparable to DreamerV2 on other tasks. We have updated line 263 in the main paper accordingly.
> > Interpretations of the notation (Equation 1, 2, 4) how do they fit together.
>
> We will add more details about each of the notations. We will elaborate more about the role of the transition predictor and the representation model, where the former is used to get the prior state and the latter provides the posterior state.
> Eq. 1 and Eq. 4 describe the world model components for VSG and SVSG, respectively. The world models in VSG comprises a recurrent model (Eq. 2) , and representation model / transition predictor. Whereas SVSG has a single module and there is no separate recurrent model. The Eq. 2 describes the Recurrent Model of VSG, whereas update equations for SVSG are mentioned in Eq. 5 & 6.
> > What exactly is the role of the transition predictor?
>
> While learning the policy, the trajectories are imagined in the latent space. Since observations are not available during unrolling sequences, the output from the transition predictor (which returns prior state- $\hat{z}_t$ for RSSM / VSG and $\hat{s}_t$ for SVSG) is used. In the submitted version, we have mentioned about learning of policy very briefly as we use the same policy as described in DreamerV2 architecture, and have added more details in Appendix B.
> > How many seeds were used in Figure 3? Was it 3 (inferred from the checklist). If so, why is it different to the 5 seeds used in other figures?
>
> For Fig. 3, we have used 5 seeds for each of the runs. We have added this detail in the description of Fig. 3 in the paper. We have used 3 seeds for experiments in Appendix F, G and H.
>
> > Why “variational” in VSG and SVSG?
>
> In this work, we have used the term VSG to define the whole block as shown in Fig. 2b. Similar to RSSM, the block generates both prior and posterior latent states, hence we have used the term variational in the name. The sparse gating represents the proposed gating mechanism. Hence, we came up with the names- Variational Sparse Gating (VSG) and Simple Variational Sparse Gating (SVSG).
>
> We hope we addressed most of your questions, and hope you consider updating your score.

---

> > ### Author Response · Authors · 2022-08-09
> > **Follow-up on Rebuttal**
> >
> > Dear Reviewer ANCN,
> >
> > Following your questions, we have added the baseline description of how GRU works in Appendix I, and updated the description of M block in Fig 2. In addition, we have explained the role of transition predictor, how model components fit together and elaborated on the purely stochastic component.
> >
> > Considering these changes, we would like to know if you can reassess our work. Given the limited time, we did our best to address your concerns and we hope that you are willing to increase the score accordingly. Kindly let us know if we can provide further clarifications.
> >
> > Thank you for your valuable feedback,
> >
> > The Authors

---

### Author Response · Authors · 2022-08-08
**Looking forward to reviewer feedbacks**

Many thanks for your efforts in reviewing our work! Since the discussion period is coming to an end, I encourage my fellow reviewers to read and acknowledge the responses. We are making our best efforts to address your concerns and looking forward to the feedbacks. Please feel free to let us know whether our responses address your concerns and whether you have other questions or suggestions.

---

### Meta-Review · Area_Chair_3MKX · 2022-08-21

**Recommendation:** Accept
**Confidence:** Less certain

**Metareview:**

The reviewers agreed this work is well written and the set of experiments are good. However, a general concern was that the interpretation / explanation of this method should be improved. The rebuttal seems to have addressed these points to a good degree and we urge the authors to revise the work to include the further explanations / experiments and analysis in the final version.

**Award:**

No

---

### Decision · Program_Chairs · 2022-09-14

Accept